# Morbidity burden and predictors of hospitalization among unaccompanied migrants and persons prone to statelessness in Ghana

Ebenezer Asare Aboagye[1,2]*, Dina Adei[1], Williams Agyemang-Duah[3]

1 Department of Planning, Kwame Nkrumah University of Science and Technology, Kumasi, Ghana,
2 Ghana Communication Technology University, Tesano, Accra-North, Ghana, 3 Department of Public Health Sciences, Queen's University, Kingston, Ontario, Canada

* eaasare@outlook.com

## Abstract

A proportion of global migration involves individuals migrating without parental or legal guardianship and those who face varied barriers to citizenship. Yet empirical evidence on their morbidity burdens and hospitalization particularly in Ghana and similar context remains scarce. This limits capacity for informed policy, planning, and response strategies aligned with global health targets. This study examined the burden of communicable and non-communicable diseases (NCDs), and predictors of hospitalization among these vulnerable groups. A cross-sectional survey was conducted from March 2024 to May 2024 among 481 purposively selected unaccompanied migrants and persons prone to statelessness in the Greater Kumasi Metropolitan Area and the Awutu Senya East Municipal Area. Data were analyzed using descriptive statistics (frequency, and percentages) and complementary log-log regression in Stata 14.2. Statistical significance was set at $p \leq 0.05$. The analyses revealed a higher prevalence of communicable diseases (23.3%) than NCDs (8.1%). Malaria (90%), flu/cold (30%), typhoid (27%), diabetes (33%) and asthma (21%) emerged as common health conditions with limited and condition-specific subgroup differences. Overall, 8.7% of respondents reported ever being hospitalized. Across models, frequent illness (Model 1: OR = 4.097, 95% CI: 2.056–8.163; Model 2: OR = 3.724, 95% CI: 1.830–7.576; Model 3: OR = 4.224, 95% CI: 2.002–8.913; all $p < 0.001$) and diagnosis with an NCD (Model 1: OR = 3.336, 95% CI: 1.611–6.906; Model 2: OR = 3.600, 95% CI: 1.737–7.460; Model 3: OR = 3.873, 95% CI: 1.861–8.058; all $p \leq 0.001$) were consistently associated with higher odds of hospitalization. These findings offer contextually bounded insights highlighting that health vulnerability among these populations is manifested less through differential disease prevalence but more through recurrent illness and NCD diagnosis necessitating hospitalization. This underscores the need for early detection, continuous care, and effective outpatient NCD management.

**Data availability statement:** The datasets used and/or analyzed during the present study cannot be shared publicly because of 1) ethical restrictions with the protocol approved by the ethics board for the study and 2) the fact that it is coming from a broader PhD study which as per the requirements future publications are expected from it to meet the PhD requirements. The data can, however, be made available upon reasonable written request. The contact details of the ethics committee that approved the conduct of this study are as follows: Committee on Human Research, Publication, and Ethics (CHRPE), School of Medicine and Dentistry, KNUST, University Post Office, Kumasi, Ghana. Email: chrpe.knust.kath@gmail.com/chrpe@knust.edu.gh.

**Funding:** The authors received no specific funding for this work.

**Competing interests:** The authors have declared that no competing interests exist.

**Abbreviations:** ASEMA, Awutu Senya East Municipal Area; BM, Behavioral Model; CHPS, Community-based Health Planning and Services; CHRPE, Committee on Human Research Publication Ethics; CLLR, Complementary Log Log Regression; GHS, Ghana Health Services; GKMA, Greater Kumasi Metropolitan Area; KNUST, Kwame Nkrumah University of Science and Technology; MoH, Ministry of Health; NDPC, National Development Planning Commission; NGO, Non-Governmental Organization; NHIS, National Health Insurance Scheme; SDG, Sustainable Development Goal; UNCHR, United Nations High Commissioner for Refugees; WHO, World Health Organization.

## Introduction

Global migration has been transformed by climate change, economic fragility, political unrest, and human rights abuses, with the effects evident in the intensification of both voluntary and forced movements within and across countries [1–6]. Of great concern is that a proportion of these movements involves minors migrating without parental or legal guardianship and people who lack recognized nationality under the operation of the laws of any country [1,7,8]. Constrained by precarious living conditions, economic instability, restricted access to formal health systems, these populations experience structurally mediated health disadvantages that heighten exposure to preventable illness [9–12].

Evidence shows that migrants morbidity spans both communicable diseases such as tuberculosis, malaria, and cholera and non-communicable diseases (NCDs) including diabetes, hypertension, and asthma [13–15]. Beyond exposure, barriers to timely and appropriate care further intensify these risks. Consistently, delayed care-seeking, treatment interruption, and avoidable hospitalization driven by documentation insecurity, financial constraints, and perceived discrimination within health systems have been documented [1,14,16,17]. Existing studies including those by Aljadeeah et al [18], Osman et al [19], Boakye et al [20], Scales et al [21], Allegri et al [22], Vignier et al [23], Chavan et al [24], and Dalmau-Bueno et al [25] have examined these patterns among the general migrant population and specific subgroups such as undocumented migrants, asylum seekers, and older adults.

Studies from the European context, for example, indicate that undocumented migrants experience approximately 19% higher risk of hospitalization for chronic conditions, 65% higher risk for acute illnesses, and more than double the risk for vaccine-preventable diseases compared with host populations [22]. Across diverse contexts, intersecting structural and institutional constraints have been documented to translate into measurable morbidity disparities, underscoring how legal precarity and social marginalization operate as fundamental determinants of health in displaced and nationality-insecure populations [18,19,21,23–25].

In Ghana, existing evidence indicates that hospitalization is shaped by chronic illness, functional limitation, and demographic factors, yet socio-economically disadvantaged groups are less likely to be admitted despite elevated need [20,26,27]. However, this evidence is confined to specific populations including adults,, and diabetic patients, leaving the morbidity burden and predictors of hospitalization among other marginalized groups such as unaccompanied migrants and persons prone to statelessness unexplored.

Unaccompanied migrants refer to minors migrating without parental or legal guardianship whilst persons prone to statelessness refer to individuals who do not have recognized nationality under the operation of the laws of any country [1,7,8]. This gap constrains progress towards Sustainable Development Goal three (SDG 3), particularly targets 3.3 (infectious diseases), 3.4 (non-communicable diseases) and 3.8 (universal health coverage). This study addresses this gap by assessing the morbidity burden and predictors of hospitalization among unaccompanied migrants and

persons prone to statelessness in Ghana. Practically, the findings are intended to inform actions toward advancing SDG 3 and its related targets in Ghana and comparable contexts.

## Methods

### Ethics statement

This study received ethical approval from the Committee on Human Research, Publication, and Ethics (CHRPE) at the School of Medical Sciences, Kwame Nkrumah University of Science and Technology (KNUST), Kumasi, Ghana (Reference: CHRPE/AP/058/24). Essential ethical safeguards upheld in the conduct of this study included confidentiality, voluntary participation, and informed consent procedures in accordance with CHRPE's guidelines on research involving human subjects. Participants were informed about the study's objectives, the voluntary nature of their participation, and autonomy to discontinue or decline answering questions deemed sensitive without any repercussions prior to the commencement of the exercise. Depending on contextual appropriateness and literacy level, written or verbal consent was obtained directly from non-minors. For minors under the care of relatives or guardians, consent was obtained from their designated caregivers. Verbal consent was specifically obtained from participants unable to provide written consent. All verbal consents were witnessed and documented in the field assistant's consent log.

### Inclusivity in global research

Additional information regarding the ethical, cultural, and scientific considerations specific to inclusivity in global research is included in the Supporting Information (S1 Checklist).

### Data and methods

This study employed data from a cross-sectional survey conducted from March 2024 to May 2024, among unaccompanied migrants and individuals prone to statelessness residing in the Greater Kumasi Metropolitan Area (GKMA) and the Awutu Senya East Municipal Area (ASEMA). The unique characteristics and relevance of the study sites to migrants and vulnerable populations are detailed elsewhere [28]. These characteristics made the sites analytically relevant for examining morbidity and hospitalization patterns in the target population. A multi-stage sampling technique was employed. First, districts with concentrations of the study population were purposefully selected informed by existing literature [29–32]. Within the purposively selected districts, all individuals who met the predefined inclusion criteria and consented to participate were enrolled. Participant selection was therefore non-random but prioritized inclusivity and feasibility.

Operationally, eligible participants included (i) individuals below 18 years who had lived outside their place of birth for more than six months under the care of persons other than a biological parent or legally recognized guardian, in line with Section 47 of the Children's Act [33] categorized as unaccompanied migrants, and (ii) individuals resident in Ghana for over five years, as stipulated in the Citizens Act [34], who are without documentary evidence of a Ghanaian or foreign nationality and are unable to reasonably establish entitlement to Ghanaian citizenship through parentage, birth registration, or naturalization pathways were classified as persons prone to statelessness. Individuals who were undocumented but could demonstrate a plausible entitlement to nationality through parentage were excluded from this category. This approach is consistent with UNHCR guidance, which emphasizes that determination of statelessness requires examination of the nationality laws of relevant States and their application in practice [35].

The adjusted target sample size was 444, calculated using Cochran's formula with a 15% allowance for non-response, 95% desired confidence level, a 0.5 conservative estimated population proportion as posited by Lwanga and Lemeshow [36] and a 0.05 margin of error. A visual presentation of participant screening, eligibility, and final inclusion is provided in **Fig 1**. Ultimately, data were obtained from 481 respondents using structured questionnaires developed for this study (see **S1 Text**). The instrument captured demographic and socioeconomic characteristics, along with information on morbidity

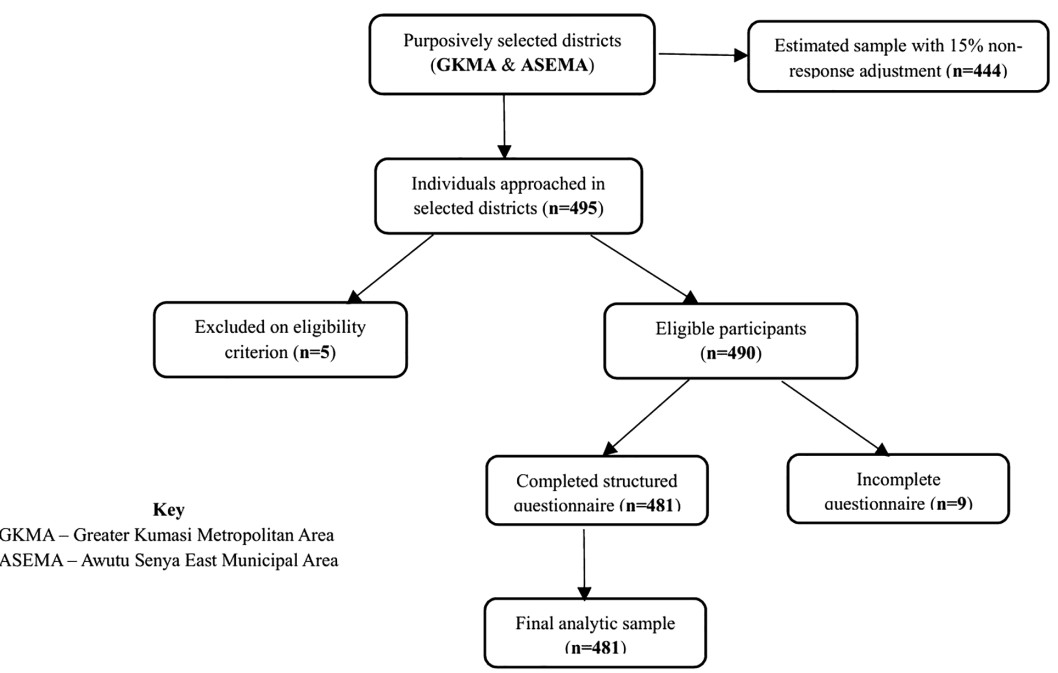

**Fig 1. Participants sampling flowchart.**

and hospitalization. Data were digitally collected with Kobo Collect on smart devices, enabling real-time entry and verification.

A pilot survey conducted in the New Juaben South Municipality (NJSM) tested the clarity, reliability, and contextual appropriateness of the instrument. NJSM was selected for the pretesting exercise due to its sociodemographic and health system comparability to the study sites. Insights from this exercise informed revisions including wording and question sequence. To accommodate linguistic diversity, the questionnaire was translated into Twi, and interpreters trusted by participants were engaged where necessary. Interviews lasted 30–50 minutes.

Data collection was carried out by six trained field assistants, all with tertiary-level education. Training which was facilitated by the principal investigator focused on the study objectives, inclusion and exclusion criteria, ethical considerations, survey administration, confidentiality, and strategies for maximizing participation. Practical sessions, including role-plays and mock interviews, were conducted to ensure competence and consistency. Fieldwork was supervised by the principal investigator and monitored by the research team to ensure data quality.

## Theoretical framework

This study employed Andersen's Behavioral Model of Health Services Use (BM) to examine predictors of hospitalization among the study population. Since its introduction in 1968, the model has remained a seminal framework for analyzing healthcare access and utilization [37–41]. This model organizes determinants into three domains, namely, predisposing, enabling, and need factors [42,43]. Predisposing factors reflect demographic attributes such as age, gender, marital status, education, and religion [44]. Enabling factors refer to the financial and logistical resources that facilitate healthcare access [43]. In Ghana, hospitalization is often contingent on health insurance [37,38], financial capacity [38,45], and proximity to healthcare facilities [46]. Yet documentation requirements for insurance registration and the burden of out-of-pocket payment systematically disadvantage vulnerable groups [38,47]. Among unaccompanied migrants and persons

prone to statelessness, language barriers may additionally reduce health literacy and awareness of available services, further restricting access to preventive and primary care. Need factors on the other hand capture both self-perceived health status and clinically assessed illness for which medical attention is required [42]. Among vulnerable populations, economic survival strategies may often delay symptom recognition and healthcare use. Many may engage in physically demanding informal labor, normalizing chronic pain and untreated symptoms, which reduces their likelihood of timely hospital visits. Consequently, hospitalizations may occur only under emergency conditions, leading to prolonged stays, higher treatment costs, and poorer health outcomes. The model's relevance in the Ghanaian context has been demonstrated in studies by Kumah et al [37], Sekyi et al [38] and Braimah et al [39]. In this study, the model guided the selection of predictor variables and the formulation of the following hypotheses: (1) need factors (frequency of illness and disease diagnosis) would significantly explain higher variation in hospitalization among unaccompanied migrants and persons prone to statelessness; (2) predisposing factors (respondent status, sex, and marital status) would significantly predict hospitalization after adjusting for need-related factors; and (3) enabling factors (district, insurance coverage, and employment status) would contribute additional explanatory power to hospitalization after accounting for need and predisposing factors.

## Measures

The dependent variable was hospitalization, defined as admission for inpatient care at a formal healthcare facility. Operationally, respondents were asked whether they had spent more than one full week on admission in a health facility. Responses were coded as binary (0 = No, 1 = Yes). Although a ≥ 24-hour threshold is conventionally employed in defining hospitalization, self-reported healthcare utilization varies in accuracy according to population characteristics, recall period, and event salience [48]. In vulnerable populations, brief inpatient stays are particularly prone to recall decay and misclassification compared to longer admissions [49,50]. The > 1-week threshold was therefore purposefully used to capture health conditions necessitating extended medical treatment or management and to improve recall and classification accuracy. Independent variables were structured in accordance with BM. However, guided by Kumah et al [37], Sekyi et al [38] and Braimah et al [39], only theoretically salient predictors were considered.

Predisposing factors included respondent status (1 = unaccompanied migrant, 2 = prone to statelessness, 3 = both), sex (1 = male, 2 = female), marital status (recoded 1 = single, 2 = married, 3 = divorced/widowed), religion (recoded 1 = Christianity, 2 = Islam, 3 = Others) to capture dominant faith-based affiliations, formal education (recoded 1 = none, 2 = primary-SHS, 3 = tertiary) to reflect functional educational attainment, and age (recoded 1 = ≤ 24, 2 = > 24). This categorization of age was adapted from the WHO demographic classification [51] and life course transitions [52,53].

Enabling factors included health insurance coverage (1 = insured, 0 = uninsured), employment status (recoded 1 = employed, 2 = student, 3 = unemployed/retired) to distinguish economic participation from dependency, district of residence (1 = GKMA, 2 = ASEMA), locality of residence (recoded 1 = rural/peri-urban/slum, 2 = urban) to capture differences in service access between urban and non-urban settings, income and social network. Income (in Ghana cedis) was recoded 1 = ≤971 and 2 = >971 to reduce sparse cells. This threshold was chosen to align with the statutory minimum wage at the time (GH¢18.15/day; GH¢544.50/month; US$1 = GH¢12.67) and to distinguish between respondents at or below subsistence income and those above. Social network denoting individuals with whom respondents maintain close contact or share living arrangements was recoded (1 = blood relation, 2 = non-blood relation, 3 = self/alone) to reflect forms of social support. This variable was included to account for the role of social ties in resource mobilization, information exchange, resilience building and barrier navigation [54–57].

Need factors captured self-reported illness frequency, NCDs, and communicable diseases. Illness frequency in the past six months was measured on a five-point scale (1 = never, 2 = less frequent, 3 = frequent, 4 = very frequent, 5 = every day) but recoded into 'infrequent' (≤2) and 'frequent' (>2) to enhance model stability. Incidence of NCDs and communicable diseases were binary (0 = No, 1 = Yes) and based solely on respondents' responses. Detailed measurement and coding of the study's variables are provided in the supplementary file (see **S1 Text**).

                          

## Analytic framework

Data analysis was conducted using Stata 14.2. Both descriptive and inferential techniques were applied to examine prevalent illnesses, frequency of hospital admissions, and predictors of hospitalization. Descriptive statistics, including frequencies and percentages, were used to summarize sample characteristics, illness types, and hospitalization patterns. To address the primary objective, proportions were computed to estimate disease prevalence and cross-tabulations were performed to examine morbidity patterns across respondent characteristics. Statistical significance of observed differences was assessed using Pearson's chi-square tests of independence (or Fisher's exact tests where cell counts were small) at a 5% significance level. For predictors of hospitalization, inferential analysis employed complementary log-log regression (CLLR). This approach was appropriate given the binary nature of the outcome variable (0 = No, 1 = Yes) and the relative rarity of hospitalization events, for which CLLR is recommended [26].

Three regression models were specified to systematically assess the effects of predictors. In the base model, need variables were included. In the second model, predisposing factors were added. In the final model, enabling factors were added. This structure was adopted to assess the effect of the study's primary predictors (need factors) controlling for the incremental explanatory effects of predisposing and enabling confounders, respectively. To enhance model stability and interpretability, univariate analysis (see **S1 Table**) with hospitalization as outcome variable was estimated to assess crude associations between predictors and the outcome variable. Based on the analysis, significant predictors (p ≤ 0.05) were considered candidates for inclusion in the multivariate analysis. Additionally, predisposing variables were deliberately minimized to include only respondent status, sex, and marital status. Respondent status is central to the study as it distinguishes between unaccompanied migrants, persons prone to statelessness, and those with intersecting vulnerabilities, thereby directly shaping exposure to structural exclusion and health risks [58].

Sex was included given consistent evidence of sex-based disparities in access and use of health service, particularly in contexts where cultural norms constrain women's healthcare access [59]. Marital status was also retained, as social support derived from marriage/cohabitation or its absence has been shown to significantly influence health behaviour globally [60]. Among enabling factors, health insurance coverage, employment status, and district of residence were prioritized. Health insurance coverage is a decisive determinant of healthcare seeking behaviour in low- and middle-income countries, where out-of-pocket payments remain a barrier to care [38,47].

Employment status reflects both economic capacity and access to financial resources, which is a common barrier to healthcare seeking in Ghana and similar context [45]. District of residence was retained to capture geographical inequities in healthcare infrastructure, as spatial disparities in service distribution are a persistent determinant of healthcare use in Ghana and similar developing contexts [46,61]. Associations were reported using odds ratios (OR), with statistical significance set at p ≤ 0.05. Model fit was evaluated using Wald's Chi-square to assess the explanatory contribution of each model.

## Results

Table 1 presents the descriptive characteristics of the study population. Hospitalization was reported by 8.7% of the respondents. The sample was predominately composed of persons prone to statelessness (61.5%), individuals under 25 years (55.7%), male (61.7%), and single (76.1%). Additionally, the majority had primary to secondary education (76.5%), earned below GH₵971 (*1 dollar = 12.67 cedis at the time of data collection*) (92.1%), and without health insurance (64.4%). In relation to morbidity, 23.3% reported being diagnosed with a communicable disease, while 8.1% reported an NCD. In relation to frequency of illness, majority reported infrequent illness (91.1%).

Distribution of reported NCDs is presented in Table 2. Among the reported NCD conditions, diabetes prevalence was 33%. Diabetes prevalence was significant among persons prone to statelessness and GKMA residents, with no significant differences by sex, age, or locality. Asthma with an overall prevalence of 21% was prevalent among unaccompanied migrants and younger respondents (≤24 years). Eye problems (15%) were significantly common among rural/peri-urban/

**Table 1. Descriptive statistics.**

| Themes | Variables | Response | Frequency | Percent (%) |
|---|---|---|---|---|
| Outcome | Hospitalization | No | 439 | 91.3 |
| | | Yes | 42 | 8.7 |
| Predisposing | Respondent Status | Unaccompanied Migrant | 112 | 23.3 |
| | | Prone to Statelessness | 296 | 61.5 |
| | | Unaccompanied & Stateless | 73 | 15.2 |
| | Sex | Male | 297 | 61.7 |
| | | Female | 184 | 38.3 |
| | Marital Status | Single | 366 | 76.1 |
| | | Married | 95 | 19.8 |
| | | Divorced/Widowed | 20 | 4.2 |
| | Age | ≤ 24 | 268 | 55.7 |
| | | > 24 | 213 | 44.3 |
| | Religion | Christianity | 279 | 58.0 |
| | | Islam | 155 | 32.2 |
| | | Others | 47 | 9.8 |
| | Formal education | None | 80 | 16.6 |
| | | Primary-SHS | 368 | 76.5 |
| | | Tertiary | 33 | 6.9 |
| Enabling | Employment Status | Employed | 131 | 27.2 |
| | | Student | 174 | 36.2 |
| | | Unemployed/Retired | 176 | 36.6 |
| | Health Insurance | Uninsured | 310 | 64.4 |
| | | Insured | 171 | 35.6 |
| | District of Residence | GKMA | 231 | 48.0 |
| | | ASEMA | 250 | 52.0 |
| | Income | ≤ 971 | 443 | 92.1 |
| | | > 971 | 38 | 7.9 |
| | Locality | Rural/Peri-Urban/Slum | 245 | 50.9 |
| | | Urban | 236 | 49.1 |
| | Social network | Blood Relation | 176 | 36.6 |
| | | Non-Blood Relation | 197 | 41.0 |
| | | Alone | 108 | 22.5 |
| Need | Frequency of illness | Infrequent | 438 | 91.1 |
| | | Frequent | 43 | 8.9 |
| | NCDs | No | 442 | 91.9 |
| | | Yes | 39 | 8.1 |
| | Communicable Disease | No | 369 | 76.7 |
| | | Yes | 112 | 23.3 |

slum residents, while ear problems (15%) were significant among ASEMA residents. Hypertension (13%) showed a significant association with urban residence only. All remaining conditions occurred at extremely low prevalence (≤5%), with no statistically significant associations.

**Table 2.** Prevalence and sociodemographic associations of NCD among respondents (N = 39).

| Condition | Prevalence n (%) | 95% CI | Explanatory Variable | Test Statistic (df) | p-value |
|---|---|---|---|---|---|
| Diabetes | 13 (33) | 0.199 - 0.501 | Respondent status | χ²(2) = 6.53 | 0.038 |
| | | | Age group | χ²(1) = 3.28 | 0.070 |
| | | | Sex | χ²(1) = 0.52 | 0.819 |
| | | | District of residence | χ²(1) = 7.43 | 0.006 |
| | | | Locality of residence | χ²(1) = 0.848 | 0.357 |
| Asthma | 8 (21) | 0.103 - 0.368 | Respondent status | χ²(2) = 7.90 | 0.019 |
| | | | Age group | χ²(1) = 5.28 | 0.022 |
| | | | Sex | χ²(1) = 0.17 | 0.682 |
| | | | District of residence | χ²(1) = 3.37 | 0.066 |
| | | | Locality of residence | χ²(1) = 1.92 | 0.166 |
| Eye Problem | 6 (15) | 0.068 - 0.311 | Respondent status | χ²(2) = 1.81 | 0.404 |
| | | | Age group | χ²(1) = 0.00 | 0.946 |
| | | | Sex | χ²(1) = 1.54 | 0.215 |
| | | | District of residence | χ²(1) = 0.47 | 0.493 |
| | | | Locality of residence | χ²(1) = 5.25 | 0.022 |
| Ear problems | 6 (15) | 0.068 - 0.311 | Respondent status | χ²(2) = 4.55 | 0.103 |
| | | | Age group | χ²(1) = 2.92 | 0.088 |
| | | | Sex | χ²(1) = 0.30 | 0.058 |
| | | | District of residence | χ²(1) = 8.27 | 0.004 |
| | | | Locality of residence | χ²(1) = 0.17 | 0.677 |
| Hypertension | 5 (13) | 0.052 - 0.282 | Respondent status | χ²(2) = 1.52 | 0.468 |
| | | | Age group | χ²(1) = 2.25 | 0.134 |
| | | | Sex | χ²(1) = 0.03 | 0.864 |
| | | | District of residence | χ²(1) = 1.58 | 0.209 |
| | | | Locality of residence | χ²(1) = 3.99 | 0.046 |
| Insomnia | 2 (5) | 0.012 - 0.193 | Respondent status | χ²(2) = 1.63 | 0.443 |
| | | | Age group | χ²(1) = 0.00 | 0.970 |
| | | | Sex | χ²(1) = 0.04 | 0.851 |
| | | | District of residence | χ²(1) = 1.81 | 0.179 |
| | | | Locality of residence | χ²(1) = 0.70 | 0.791 |
| Mental disorder | 1 (3) | 0.003 - 0.174 | Respondent status | χ²(2) = 6.98 | 0.031 |
| | | | Age group | χ²(1) = 0.98 | 0.323 |
| | | | Sex | χ²(1) = 0.79 | 0.373 |
| | | | District of residence | χ²(1) = 0.88 | 0.348 |
| | | | Locality of residence | χ²(1) = 0.71 | 0.398 |
| Stroke | 1 (3) | 0.003 - 0.174 | Respondent status | χ²(2) = 0.79 | 0.673 |
| | | | Age group | χ²(1) = 1.08 | 0.299 |
| | | | Sex | χ²(1) = 0.79 | 0.373 |
| | | | District of residence | χ²(1) = 0.88 | 0.348 |
| | | | Locality of residence | χ²(1) = 0.71 | 0.398 |
| Depression | 1 (3) | 0.003 - 0.174 | Respondent status | χ²(2) = 2.31 | 0.315 |
| | | | Age group | χ²(1) = 0.98 | 0.323 |
| | | | Sex | χ²(1) = 1.33 | 0.249 |
| | | | District of residence | χ²(1) = 1.20 | 0.274 |
| | | | Locality of residence | χ²(1) = 1.48 | 0.225 |

*(Continued)*

**Table 2.** (Continued)

| Condition | Prevalence n (%) | 95% CI | Explanatory Variable | Test Statistic (df) | p-value |
|---|---|---|---|---|---|
| Arthritis | 1 (3) | 0.003 - 0.174 | Respondent status | $\chi^2(2) = 0.79$ | 0.673 |
| | | | Age group | $\chi^2(1) = 1.08$ | 0.299 |
| | | | Sex | $\chi^2(1) = 0.79$ | 0.373 |
| | | | District of residence | $\chi^2(1) = 1.20$ | 0.274 |
| | | | Locality of residence | $\chi^2(1) = 1.48$ | 0.225 |
| Cancer | No cases observed | | | | |
| Kidney disease | No cases observed | | | | |

**Notes**

1. Prevalence is reported as number (percentage) of respondents diagnosed with each condition.

2. Conditions with zero prevalence were excluded from inferential analysis and are presented descriptively.

3. Given the sample size, findings should be interpreted as exploratory.

Distribution of reported communicable diseases is presented in Table 3. Among communicable diseases reported, malaria was the most prevalent condition (90%), with no statistically significant variation across respondent status, age, sex, district, or locality. Flu/cold (30%) and typhoid (27%) were also common conditions. Typhoid prevalence was significant among ASEMA residents, while flu/cold showed no significant subgroup differences. Cholera (8%) was significantly common among younger respondents (≤24 years), females, and ASEMA residents. Measles/chickenpox (6%) was significantly associated with dual vulnerability status, while STIs/STDs (4%) were significant among GKMA and urban residents. Yellow fever was rare (3%) with no statistically significant associations.

Table 4 summarizes the predictors of inpatient care. In Model 1, frequent illness, and diagnosis with an NCD were both significant predictors of hospitalization. Succinctly, respondents reporting frequent illness had over four times higher odds of hospitalization compared with those reporting infrequent illness (OR = 4.097; 95% CI: 2.056–8.163; p < 0.001). Similarly, respondents diagnosed with an NCD had over three times higher odds of hospitalization than those without such a diagnosis (OR = 3.336; 95% CI: 1.611–6.906; p = 0.001). Diagnosis of a communicable disease was not statistically associated with hospitalization.

In Model 2, after adjusting for demographic characteristics, the associations for frequent illness (OR = 3.724; 95% CI: 1.830–7.576; p < 0.001) and NCD diagnosis (OR = 3.600; 95% CI: 1.737–7.460; p = 0.001) remained statistically significant with minimal attenuation. Additionally, individuals prone to statelessness exhibited significantly lower odds of hospitalization compared with unaccompanied migrants (OR = 0.385; 95% CI: 0.163–0.909; p = 0.030). Furthermore, divorced, or widowed respondents exhibited over four times higher odds of hospitalization compared with single respondents (OR = 4.042; 95% CI: 1.278–12.784; p = 0.017). Sex and communicable disease diagnosis were not statistically significant in this model.

In the final model which further adjusted for contextual and healthcare access factors, frequent illness (OR = 4.224; 95% CI: 2.002–8.913; p < 0.001) and NCD diagnosis (OR = 3.873; 95% CI: 1.861–8.058; p < 0.001) remained robust predictors of hospitalization. The previously observed association between being prone to statelessness and lower odds of hospitalization was attenuated (OR = 0.558; 95% CI: 0.188–1.652; p = 0.292). However, female respondents demonstrated significantly lower odds of hospitalization compared with males (OR = 0.469; 95% CI: 0.233–0.946; p = 0.035). Divorced or widowed status remained associated with over four times higher odds of hospitalization (OR = 4.808; 95% CI: 1.423–16.247; p = 0.011). District of residence, insurance, and employment status were not significantly associated with hospitalization. Across models, likelihood ratio tests indicated progressive improvement in model fit, with the final model demonstrating the best overall fit (Wald $\chi^2 = 55.58$; p < 0.001).

**Table 3. Prevalence and sociodemographic associations of communicable diseases among respondents (N = 112).**

| Condition | Prevalence n (%) | 95% CI | Explanatory Variable | Test Statistic (df) | p-value |
|---|---|---|---|---|---|
| Malaria | 101 (90) | 0.830 - 0.945 | Respondent status | $\chi^2(2) = 2.73$ | 0.255 |
| | | | Age group | $\chi^2(1) = 0.26$ | 0.613 |
| | | | Sex | $\chi^2(1) = 0.24$ | 0.623 |
| | | | District of residence | $\chi^2(1) = 3.04$ | 0.081 |
| | | | Locality of residence | $\chi^2(1) = 0.00$ | 0.995 |
| Flu/Cold | 34 (30) | 0.224 - 0.396 | Respondent status | $\chi^2(2) = 1.86$ | 0.395 |
| | | | Age group | $\chi^2(1) = 2.59$ | 0.107 |
| | | | Sex | $\chi^2(1) = 3.76$ | 0.053 |
| | | | District of residence | $\chi^2(1) = 0.75$ | 0.386 |
| | | | Locality of residence | $\chi^2(1) = 3.42$ | 0.064 |
| Typhoid | 30 (27) | 0.193 - 0.359 | Respondent status | $\chi^2(2) = 2.10$ | 0.350 |
| | | | Age group | $\chi^2(1) = 0.12$ | 0.731 |
| | | | Sex | $\chi^2(1) = 0.28$ | 0.595 |
| | | | District of residence | $\chi^2(1) = 9.67$ | 0.002 |
| | | | Locality of residence | $\chi^2(1) = 0.51$ | 0.477 |
| Cholera | 9 (8) | 0.042 - 0.149 | Respondent status | $\chi^2(2) = 1.79$ | 0.408 |
| | | | Age group | $\chi^2(1) = 8.79$ | 0.003 |
| | | | Sex | $\chi^2(1) = 4.68$ | 0.031 |
| | | | District of residence | $\chi^2(1) = 5.29$ | 0.021 |
| | | | Locality of residence | $\chi^2(1) = 1.76$ | 0.184 |
| Measles/ Chickenpox | 7 (6) | 0.030 - 0.127 | Respondent status | $\chi^2(2) = 10.60$ | 0.005 |
| | | | Age group | $\chi^2(1) = 3.27$ | 0.071 |
| | | | Sex | $\chi^2(1) = 1.80$ | 0.180 |
| | | | District of residence | $\chi^2(1) = 2.37$ | 0.124 |
| | | | Locality of residence | $\chi^2(1) = 0.02$ | 0.883 |
| STI/STDs | 4 (4) | 0.013 - 0.093 | Respondent status | $\chi^2(2) = 2.31$ | 0.316 |
| | | | Age group | $\chi^2(1) = 1.28$ | 0.259 |
| | | | Sex | $\chi^2(1) = 2.13$ | 0.144 |
| | | | District of residence | $\chi^2(1) = 13.06$ | 0.000 |
| | | | Locality of residence | $\chi^2(1) = 4.96$ | 0.026 |
| Yellow Fever | 3 (3) | 0.009 - 0.081 | Respondent status | $\chi^2(2) = 5.87$ | 0.053 |
| | | | Age group | $\chi^2(1) = 2.78$ | 0.096 |
| | | | Sex | $\chi^2(1) = 1.47$ | 0.225 |
| | | | District of residence | $\chi^2(1) = 0.98$ | 0.322 |
| | | | Locality of residence | $\chi^2(1) = 0.19$ | 0.667 |
| Tuberculosis | No cases observed | | | | |
| Hepatitis | No cases observed | | | | |

**Notes**

1. Prevalence is reported as number (percentage) of respondents diagnosed with each condition.

2. Conditions with zero prevalence were excluded from inferential analysis and are presented descriptively.

3. Given the sample size, findings should be interpreted as exploratory.

**Table 4. Predictors of hospitalization.**

| Parameter | Model 1 | | Model 2 | | Final Model (3) | |
|---|---|---|---|---|---|---|
| | OR (95% CI) | p-value | OR (95% CI) | p-value | OR (95% CI) | p-value |
| **Frequency of illness** | | | | | | |
| Infrequent (Ref) | | | | | | |
| Frequent | 4.097* (2.056-8.163) | 0.000 | 3.724* (1.830-7.576) | 0.000 | 4.224* (2.002-8.913) | 0.000 |
| **NCDs** | | | | | | |
| No (Ref) | | | | | | |
| Yes | 3.336* (1.611-6.906) | 0.001 | 3.600* (1.737-7.460) | 0.001 | 3.873* (1.861-8.058) | 0.000 |
| **Communicable Disease** | | | | | | |
| No (Ref) | | | | | | |
| Yes | 1.756 (0.920-3.353) | 0.088 | 1.771 (0.924-3.394) | 0.085 | 1.335 (0.659-2.706) | 0.423 |
| **Respondent status** | | | | | | |
| Unaccompanied Migrant (Ref) | | | | | | |
| Prone to Statelessness | | | 0.385* (0.163-0.909) | 0.030 | 0.558 (0.188-1.652) | 0.292 |
| Unaccompanied & Stateless | | | 1.003 (0.415-2.426) | 0.994 | 1.477 (0.574-3.804) | 0.419 |
| **Sex** | | | | | | |
| Male (Ref) | | | | | | |
| Female | | | 0.516 (0.258-1.030) | 0.061 | 0.469* (0.233-0.946) | 0.035 |
| **Marital Status** | | | | | | |
| Single (Ref) | | | | | | |
| Married | | | 1.245 (0.433-3.581) | 0.684 | 1.301 (0.437-3.872) | 0.636 |
| Divorced/Widowed | | | 4.042* (1.278-12.784) | 0.017 | 4.808* (1.423-16.247) | 0.011 |
| **District** | | | | | | |
| GKMA (Ref) | | | | | | |
| ASEMA | | | | | 1.790 (0.737-4.342) | 0.198 |
| **Insurance Coverage** | | | | | | |
| Uninsured (Ref) | | | | | | |
| Insured | | | | | 1.676 (0.838-3.351) | 0.144 |
| **Employment Status** | | | | | | |
| Employed (Ref) | | | | | | |
| Student | | | | | 0.721 (0.208-2.506) | 0.607 |
| Unemployed/Retired | | | | | 0.416 (0.155-1.111) | 0.080 |
| **(Intercept)** | 0.050 (0.032-0.077) | 0.000 | 0.090 (0.046-0.177) | 0.000 | 0.062 (0.017-0.220) | 0.000 |
| Model Fitness | | | | | | |

*(Continued)*

**Table 4.** (Continued)

| Parameter | Model 1 | | Model 2 | | Final Model (3) | |
|---|---|---|---|---|---|---|
| | OR (95% CI) | p-value | OR (95% CI) | p-value | OR (95% CI) | p-value |
| Wald's Chi-Square (p-value) | 36.57 (0.000) | | 47.69 (0.000) | | 55.58 (0.000) | |

**Note**

1. Model 1 includes need variables only (frequency of illness, NCD and communicable illness diagnosis).

2. Model 2 adjusts Model 1 for predisposing factors (respondent status, gender, marital status).

3. Model 3 further adjusts Model 2 for enabling factors (district of residence, insurance coverage, employment status).

4. Statistical significance set at $p < 0.05$.

5. OR > 1 indicates higher odds of hospitalization; OR < 1 indicates lower odds.

6. 42 respondents (8.7%) reported hospitalization while 439 (91.3%) reported no hospitalization.

## Discussion

This study provides a novel assessment of morbidity burden and hospitalization among unaccompanied migrants and persons prone to statelessness in Ghana. The results indicate a higher prevalence of communicable diseases than NCDs. Malaria, flu/cold, typhoid, diabetes and asthma emerged as the most common conditions. Despite this morbidity profile, hospitalization was rare (8.7%). Across all the models, frequent illness, and diagnosis with an NCD were significantly associated with higher odds of hospitalization.

Morbidity prevalence in this study was modest, particularly for NCD, diverging from international evidence which documents elevated disease burdens among migrant populations due to socioeconomic constraints and restricted healthcare access [22,23,62–67]. In the Ghanaian context, where population-based studies report higher NCD prevalence [25], the comparatively low rates observed here are more plausibly interpreted as reflecting limited detection within marginalized populations rather than a lower underlying disease burden. This interpretation is consistent with existing literature documenting persistent barriers to formal healthcare access among uninsured and socioeconomically vulnerable groups including migrants, refugees, older adult populations and persons prone to statelessness [9–12,26,68,69].

Within the observed morbidity profiles, diabetes emerged as the most prevalent NCD, aligning with national evidence of Ghana's epidemiological transition [70–73]. Its higher prevalence among persons prone to statelessness and urban residents is consistent with evidence that place-based and socioeconomic contexts structure NCD risk. In the Ghanaian context for example, significant variations in NCD multimorbidity have been attributed to neighborhood-level factors such as income [74], with community socioeconomic conditions including level of poverty observed to be independently associated with NCD risk [75]. Similar chronic disease burdens have been documented among displaced populations such as refugees in other contexts including America, the Middle East and Asia [76,77].

Similarly, malaria's dominance among communicable diseases is consistent with national and global evidence of its persistence in underserved populations [78–80]. The concentration of typhoid in ASEMA can plausibly be attributed to spatial inequalities in environmental health systems including adequate sanitation and access to safe water. Similar patterns have been documented in comparable Ghanaian settings where typhoid, diarrhea, and other waterborne infections are major contributors to disease burden [81,82]. Importantly, most communicable and non-communicable conditions showed no statistically significant variation across key socio-demographic subgroups, suggesting that morbidity exposure in this population is broadly distributed rather than concentrated within defined demographic strata.

Across all models, frequent illness and NCD diagnosis were the dominant predictors of hospitalization. This indicates that hospitalization in this population is driven primarily by need factors as has been observed among population of older

adults [26,68,83]. This finding accords closely with the BM and further suggests that need factors exert the strongest and most proximate influence on hospitalization. In contrast, enabling factors such as health insurance coverage, employment status, and district of residence were not independently associated with hospitalization after adjustment. This suggests that structural barriers may constrain access to care earlier in the illness trajectory rather than at the point of severe morbidity requiring admission.

Finally, predisposing factors demonstrated limited and model-specific effects. While being prone to statelessness was associated with lower odds of hospitalization in the partially adjusted model, this association was attenuated after accounting for contextual and access-related variables. This indicates that documentation vulnerability does not independently shape hospitalization once need and enabling factors are considered. Sex and marital status showed independent associations only in the fully adjusted model, with females exhibiting lower odds of hospitalization and divorced or widowed respondents showing higher odds. These findings reflect differential patterns of hospitalization [26,60,68] rather than differential morbidity risk. This further reinforces the BM's proposition that predisposing characteristics conditions responses to illness rather than disease occurrence itself.

## Policy and practice implications

The findings of this study offer contextually bounded implications that warrant a cautious generalization and application. The predominance of communicable diseases, particularly malaria relative to NCDs, indicates that infectious disease prevention remains a critical public health priority within the study population. While these results are not generalizable, it supports the continued prioritization of established preventive interventions, including vector control, sanitation, and basic health education, in communities hosting socioeconomically marginalized groups. This has direct implications for adopted strategies towards the achievement of targets 3.3 and 3.4 of SDG 3.

Also, the minimal hospitalization rate and consistent association of hospitalization with illness frequency and NCD diagnosis highlight that inpatient care in this population is largely triggered by advanced or recurrent morbidity rather than routine healthcare use. This pattern highlights the potential value of strengthening early detection and outpatient management of chronic and recurrent illnesses within primary care and community health systems serving vulnerable groups. Importantly, the absence of independent associations between hospitalization and enabling factors such as health insurance enrolment or employment status cautions against over-interpreting structural access variables as direct predictors of hospitalization among socioeconomically marginalized groups.

Finally, the limited and model-dependent associations observed for sex and marital status indicate that predisposing characteristics may shape responses to illness once hospitalization becomes necessary, rather than influencing morbidity risk per se. These suggest that service delivery approaches should remain attentive to heterogeneity in hospitalization patterns without assuming uniform vulnerability across demographic subgroups. However, given the study's cross-sectional design, self-reported measures, and geographically restricted sample, such implications should be viewed as hypothesis-generating rather than prescriptive.

## Limitations

Despite the useful insights and implications, this study is not without limitations. First, morbidity and hospitalization were self-reported, introducing potential recall bias and misclassification. Secondly, classification of persons prone to statelessness relied on documentation status and self-reported legal entitlement and may have misclassified some undocumented individuals. Furthermore, the use of an extended inpatient threshold as a measure of hospitalization improved recall but likely underestimated prevalence and may have influenced effect estimates. Moreso, the binary outcome of hospitalization yielded a markedly low success rate relative to non-hospitalization due to the restrictive threshold. This could have constrained statistical power and influenced estimates. However, this limitation was methodologically moderated through the application of CLLR with hierarchical variable entry, an analytic approach well suited for modelling rare events and

mitigating estimation bias under such conditions. Additionally, the reliance on purposive site selection and non-probability participant selection within the selected districts limits representativeness and generalizability. Again, several disease categories had small cell counts, which constrained statistical power and may have influenced regression estimates. Finally, the cross-sectional design precludes causal inference.

## Conclusion

This study provides novel insights on the morbidity patterns and predictors of hospitalization among unaccompanied migrants and persons prone to statelessness in Ghana—groups whose invisibility in data translates in invisibility in health policy, planning and response in Ghana and similar context. The findings suggests that vulnerability in these populations is manifested less through differences in disease prevalence but more through recurrent illness and NCD diagnosis necessitating hospitalization. This underscores the importance of early detection, continuity of care, and outpatient management for marginalized migrant populations, among whom unmet health needs may remain obscured until hospitalization becomes unavoidable. While the evidence does not justify definitive conclusions, it suggests that health strategies in Ghana and similar context could be strengthened by integrating targeted support for vulnerable groups within the broader primary healthcare and social protection frameworks. In the Ghanaian context, the Ministry of Health (MoH), Ghana Health Services (GHS), non-governmental organizations (NGO's), faith-based organizations, and research institutions could implement regular community-based NCD screening targeting high-risk age groups, socioeconomically marginalized populations and provide subsidized medication for detected diseases in the short-medium term.

## Supporting information

**S1 Text. Data collection instrument (questionnaire).**
(DOCX)

**S1 Table. Univariate regression results.**
(DOCX)

**S1 Checklist. Inclusivity in global research.**
(DOCX)

## Author contributions

**Conceptualization:** Ebenezer Asare Aboagye, Dina Adei.

**Data curation:** Ebenezer Asare Aboagye.

**Formal analysis:** Ebenezer Asare Aboagye, Williams Agyemang-Duah.

**Methodology:** Ebenezer Asare Aboagye.

**Supervision:** Dina Adei.

**Writing – original draft:** Ebenezer Asare Aboagye.

**Writing – review & editing:** Ebenezer Asare Aboagye, Dina Adei, Williams Agyemang-Duah.

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
