## [Decision Letter · Decision Letter 0]

26 Jan 2026

PGPH-D-25-03075

Morbidity Burden and Predictors of Hospitalization Among Unaccompanied Migrants and Persons Prone to Statelessness in Ghana

Dear Dr. Aboagye,

Thank you for submitting your manuscript to PLOS Global Public Health. After careful consideration, we feel that it has merit but does not fully meet PLOS Global Public Health’s publication criteria as it currently stands. Therefore, we invite you to submit a revised version of the manuscript that addresses the points raised during the review process.

We look forward to receiving your revised manuscript.

Kind regards,

Ryan Essex

Academic Editor

Journal Requirements:

2. Please ensure that your Ethics Statement is available in its entirety at the beginning of your Methods section, under a subheading 'Ethics Statement'.

3. In the online submission form, you indicated that “Datasets for this study are available from the corresponding author upon reasonable request.”.

3. Uploaded as supplementary information.

4. Some material included in your submission may be copyrighted. According to PLOS’s copyright policy, authors who use figures or other material (e.g., graphics, clipart, maps) from another author or copyright holder must demonstrate or obtain permission to publish this material under the Creative Commons Attribution 4.0 International (CC BY 4.0) License used by PLOS journals. Please closely review the details of PLOS’s copyright requirements here: PLOS Licenses and Copyright. If you need to request permissions from a copyright holder, you may use PLOS's Copyright Content Permission form.

Potential Copyright Issues:

a. Appendix 2 contains a logo. We are not permitted to publish this under our CC-BY 4.0 license, even with permission. We ask that you please remove or replace it.

Additional Editor Comments (if provided):

Reviewers' comments:

Reviewer's Responses to Questions

**Comments to the Author**

1. Does this manuscript meet PLOS Global Public Health’s publication criteria? Is the manuscript technically sound, and do the data support the conclusions? The manuscript must describe methodologically and ethically rigorous research with conclusions that are appropriately drawn based on the data presented.? Is the manuscript technically sound, and do the data support the conclusions? The manuscript must describe methodologically and ethically rigorous research with conclusions that are appropriately drawn based on the data presented.

Reviewer #1: Yes

Reviewer #2: Partly

2. Has the statistical analysis been performed appropriately and rigorously?

Reviewer #1: Yes

Reviewer #2: No

3. Have the authors made all data underlying the findings in their manuscript fully available (please refer to the Data Availability Statement at the start of the manuscript PDF file)?

The PLOS Data policy requires authors to make all data underlying the findings described in their manuscript fully available without restriction, with rare exception. The data should be provided as part of the manuscript or its supporting information, or deposited to a public repository. For example, in addition to summary statistics, the data points behind means, medians and variance measures should be available. If there are restrictions on publicly sharing data—e.g. participant privacy or use of data from a third party—those must be specified.requires authors to make all data underlying the findings described in their manuscript fully available without restriction, with rare exception. The data should be provided as part of the manuscript or its supporting information, or deposited to a public repository. For example, in addition to summary statistics, the data points behind means, medians and variance measures should be available. If there are restrictions on publicly sharing data—e.g. participant privacy or use of data from a third party—those must be specified.

Reviewer #1: Yes

Reviewer #2: No

4. Is the manuscript presented in an intelligible fashion and written in standard English?

Reviewer #1: Yes

Reviewer #2: Yes

Reviewer #1: This manuscript addresses an important and understudied topic: the morbidity burden and predictors of hospitalization among unaccompanied migrants and individuals prone to statelessness in Ghana. despite its relevance, the manuscript requires substantial revisions before it can be considered for publication.

• Revise and justify the definition of Statelessness using established international frameworks (e.g., UNHCR, Ghana Nationality Act). Clarify how the study distinguishes undocumented but legally entitled individuals from those genuinely at risk of statelessness.

• Hospitalization is defined as inpatient stay for more than one full week, which is inconsistent with global health services research (typically ≥24 hours). This definition excludes a large proportion of real hospitalizations, artificially depresses prevalence (8.7%), and may bias regression results. Revise the definition of Hospitalization or provide a strong methodological justification for using a 7-day threshold.

• Present a clear sampling flowchart. Justify the selection of districts and communities. Explain how randomness was ensured at the participant level.

• Provide justification for each recoding decision as Variable Recoding reduces Analytical Precision, including statistical (model stability) or theoretical reasons.

• tables 2 and 3 show no statistically significant differences for any NCD or communicable disease across demographic and geographic subgroups (all p > 0.1). Despite this, the narrative repeatedly describes patterns as “higher,” “more common,” or “notable.” This is inappropriate and misleading. Rewrite the Results section to strictly reflect statistically supported findings. Where no significant differences exist, state this clearly and avoid narrative speculation based on raw counts.

• Regression tables lack 95% confidence intervals, which are mandatory for odds ratio interpretation. Include confidence intervals for all ORs in all models.

• Interpretations across Models 1–3 are sometimes incorrect. For example, “prone to statelessness” is statistically significant in Model 2 but not in Model 3; however, the discussion presents it as a consistent predictor. Revise narrative to accurately reflect model-specific findings and note attenuation or loss of significance.

• The Discussion section restates results extensively and occasionally presents speculative explanations (e.g., lifestyle shifts, informal care behaviors) that were not measured in the study. While Andersen’s Behavioral Model is cited, its application is superficial. The discussion does not meaningfully connect predisposing, enabling, and need factors to the results. Given that most subgroup differences were non-significant, this should be a central part of the Discussion, yet it is not addressed.

• Reframe policy recommendations to align directly with supported findings and emphasize caution due to limited generalizability.

• While some limitations are acknowledged, several important ones are missing: Potential misclassification of statelessness, Highly restrictive hospitalization definition, Non-representative sampling from only two districts, Very small cell counts for diseases (1–11 cases in some categories), Instability of regression estimates due to sparse data.

Expand the Limitations section to address these substantive concerns.

• The Introduction is overly long, repetitive, and contains multiple overlapping paragraphs about migrant vulnerabilities. Reduce length by 30-40%, remove repetition, and sharpen the logical flow toward the research gap.

• Many interpretive comments appear in the Results section; others reappear in the exact same words in the Discussion. Ensure strict separation between empirical results and interpretation.

Reviewer #2: 1. In abstract, Methods section needs to be elaborated focusing on the flow of selection of these two areas along with participants, exposure and outcome assessment. Results section lacks frequency and percentages to get meaningful interpretation of the results.

2. In Introduction, the authors have claimed disproportionate health risk faced by migrant population, but statistics related to it is missing in this section. (line number 42-45)

3. In line number 78-79, what SDG goal is affected from the objective authors claims to assess?

4. In line number 83-85, the study objective doesn't inform inclusive healthcare policies and interventions. Rephrase the sentence and give the expected outcome of the study.

5. The theoretical framework given from line number 86 to 110 can be included under methodology section.

6. In the line number 119-121, the authors have claimed the areas to be distinctive for target population, need some more information of the target population distribution in these areas and what does these areas represent as in organizational hierarchy?

7. From line number 131-133, on what basis, the proportion of 0.5 was determined for sample size calculation? Sample size calculation for the determinants is missing.

8. Operational definition of Persons Prone to Statelessness, Unaccompanied Migrants is not defined explicitly.

9. In line number 202, the authors have mentioned three tier hierarchical regression analysis, what are the three tiers in that regression analysis?

10. The Inference statistics for primary objective is missing in the statistical analysis plan.

11. Information about the univariate regression and multivariable regression analysis is missing. How the variables were selected into the model is missing too.

12. The interpretation given below the table 1 is repetition of information given in the table. Mention only the relevant findings of the table 1

13. The construction of Table 2 is poor, what does the p-value indicate? is the p-value specific to one self-reported morbidity status or it is for the entire morbidity status? Frequency for each morbidity is given, percentage is missing.

95% Confidence Interval for overall prevalence for each morbidity status is missing.

14. The comment 13 applies for table 3 too.

15. In table 4, what does Infrequent illness indicate, should be given in the table. The variables included in each model should be explicitly mentioned and the descriptive statistics about the hospitalization is missing in Table 4. Only the point estimate of Odds ratio is explicitly given, 95% CI is missing.

16. Interpretation of the multivariable regression analysis for each covariates is wrong from line number 272 to 291.

**Do you want your identity to be public for this peer review?** For information about this choice, including consent withdrawal, please see our Privacy Policy..

Reviewer #1: No

Reviewer #2: No

---

## [Decision Letter · Decision Letter 1]

23 Feb 2026

PGPH-D-25-03075R1

Morbidity Burden and Predictors of Hospitalization Among Unaccompanied Migrants and Persons Prone to Statelessness in Ghana

Dear Dr. Aboagye,

Thank you for submitting your manuscript to PLOS Global Public Health. After careful consideration, we feel that it has merit but does not fully meet PLOS Global Public Health’s publication criteria as it currently stands. Therefore, we invite you to submit a revised version of the manuscript that addresses the points raised during the review process.

We look forward to receiving your revised manuscript.

Kind regards,

Ryan Essex

Academic Editor

Journal Requirements:

Additional Editor Comments (if provided):

Reviewers' comments:

Reviewer's Responses to Questions

**Comments to the Author**

Reviewer #1: (No Response)

Reviewer #2: All comments have been addressed

publication criteria? Is the manuscript technically sound, and do the data support the conclusions? The manuscript must describe methodologically and ethically rigorous research with conclusions that are appropriately drawn based on the data presented.? Is the manuscript technically sound, and do the data support the conclusions? The manuscript must describe methodologically and ethically rigorous research with conclusions that are appropriately drawn based on the data presented.

Reviewer #1: Yes

Reviewer #2: No

3. Has the statistical analysis been performed appropriately and rigorously?

Reviewer #1: Yes

Reviewer #2: No

4. Have the authors made all data underlying the findings in their manuscript fully available (please refer to the Data Availability Statement at the start of the manuscript PDF file)?

The PLOS Data policy requires authors to make all data underlying the findings described in their manuscript fully available without restriction, with rare exception. The data should be provided as part of the manuscript or its supporting information, or deposited to a public repository. For example, in addition to summary statistics, the data points behind means, medians and variance measures should be available. If there are restrictions on publicly sharing data—e.g. participant privacy or use of data from a third party—those must be specified.requires authors to make all data underlying the findings described in their manuscript fully available without restriction, with rare exception. The data should be provided as part of the manuscript or its supporting information, or deposited to a public repository. For example, in addition to summary statistics, the data points behind means, medians and variance measures should be available. If there are restrictions on publicly sharing data—e.g. participant privacy or use of data from a third party—those must be specified.

Reviewer #1: No

Reviewer #2: Yes

5. Is the manuscript presented in an intelligible fashion and written in standard English?

Reviewer #1: Yes

Reviewer #2: Yes

Reviewer #1: After reviewing the revised manuscript carefully, the following concerns remain either not fully addressed or only partially addressed:

• The definition of statelessness is grounded in the Ghana Citizenship Act, but it is not explicitly aligned with established international frameworks such as the 1954 Convention or UNHCR’s formal definition. The manuscript does not clearly situate its operational definition within global legal standards.

• The hospitalization definition (>1 week inpatient stay) is justified on recall grounds, but no sensitivity analysis using the conventional ≥24-hour definition is provided. The manuscript does not empirically demonstrate how this restrictive threshold may have influenced prevalence or regression estimates.

• A clear sampling flowchart is still missing. Although the text now clarifies purposive district selection and non-random inclusion, there is no visual presentation of participant screening, eligibility, and final inclusion.

• The use of the term “multi-stage sampling” may remain misleading because the final participant inclusion was non-probability and inclusive rather than randomized. The terminology could still imply greater methodological rigor than was actually used.

• Some residual narrative language in the Results section still uses comparative terms such as “more prevalent” or “higher” in contexts where statistical support is weak or exploratory. While improved, a few phrases could be further neutralized.

• The Discussion still contains speculative explanations (for example, sanitation or urban poverty mechanisms) that were not directly measured in the study. These interpretations are softer than before but remain somewhat inferential.

• The Introduction, although shortened, but Some repetition around migrant vulnerability remains.

Everything else, including regression interpretation corrections, confidence intervals, recoding justifications, expanded limitations, and improved model-specific interpretation, appears to have been adequately addressed.

Reviewer #2: 1. Sample size calculation for the primary objective - proportion was taken from Sample Size Determination in Health Studies: A Practical Guide by Lwanga and Lemeshow. But, the guide recommends to use 0.5 as proportion if previous published literature is not available. So, the author claim that there is no published literature related to proportion of communicable and NCD among the selected study population? Sample size calculation for predictors of hospitalization is not explicitly stated.

2. Operational definition of Persons Prone to Statelessness, Unaccompanied Migrants is not

defined explicitly.

3. The Inference statistics for primary objective is still missing in the statistical analysis plan.

4. Line number 219-234 - The authors have given rationale for variable selection in description. But, it would be better to include Directed acyclic graphs (DAG) or statistical principle of variable selection in the multivariable regression model.

5. Table 2 Prevalence and Sociodemographic Associations of NCD Among Respondents (N = 39) and Table 3 Prevalence and Sociodemographic Associations of Communicable Diseases Among Respondents (N = 112) along with their interpretation are not required as per your objective of interest. Univariate analysis should be performed with hospitalization as outcome of interest.

6. In Table 4, as per the frequency of outcome of interest i.e (hospitalization), it is advisable to include 4 to 5 variables in the model. But the authors have included more than the advisable predictor variables in the model.

**Do you want your identity to be public for this peer review?** For information about this choice, including consent withdrawal, please see our Privacy Policy..

Reviewer #1: No

Reviewer #2: No

---

## [Decision Letter · Decision Letter 2]

31 Mar 2026

Morbidity Burden and Predictors of Hospitalization Among Unaccompanied Migrants and Persons Prone to Statelessness in Ghana

PGPH-D-25-03075R2

Dear Mr Aboagye,

We are pleased to inform you that your manuscript 'Morbidity Burden and Predictors of Hospitalization Among Unaccompanied Migrants and Persons Prone to Statelessness in Ghana' has been provisionally accepted for publication in PLOS Global Public Health.

Best regards,

Ryan Essex

Academic Editor

Associate editor comments

While one reviewer has identified that further amendments be made, I believe the authors have responded appropriately in the last round of reviews and provided a rationale for changes they did not incorporate.

Reviewer Comments (if any, and for reference):

Reviewer's Responses to Questions

**Comments to the Author**

Reviewer #1: All comments have been addressed

Reviewer #2: All comments have been addressed

publication criteria? Is the manuscript technically sound, and do the data support the conclusions? The manuscript must describe methodologically and ethically rigorous research with conclusions that are appropriately drawn based on the data presented.? Is the manuscript technically sound, and do the data support the conclusions? The manuscript must describe methodologically and ethically rigorous research with conclusions that are appropriately drawn based on the data presented.

Reviewer #1: Yes

Reviewer #2: No

3. Has the statistical analysis been performed appropriately and rigorously?

Reviewer #1: I don't know

Reviewer #2: No

4. Have the authors made all data underlying the findings in their manuscript fully available (please refer to the Data Availability Statement at the start of the manuscript PDF file)?

The PLOS Data policy requires authors to make all data underlying the findings described in their manuscript fully available without restriction, with rare exception. The data should be provided as part of the manuscript or its supporting information, or deposited to a public repository. For example, in addition to summary statistics, the data points behind means, medians and variance measures should be available. If there are restrictions on publicly sharing data—e.g. participant privacy or use of data from a third party—those must be specified.requires authors to make all data underlying the findings described in their manuscript fully available without restriction, with rare exception. The data should be provided as part of the manuscript or its supporting information, or deposited to a public repository. For example, in addition to summary statistics, the data points behind means, medians and variance measures should be available. If there are restrictions on publicly sharing data—e.g. participant privacy or use of data from a third party—those must be specified.

Reviewer #1: Yes

Reviewer #2: Yes

5. Is the manuscript presented in an intelligible fashion and written in standard English?

Reviewer #1: Yes

Reviewer #2: Yes

Reviewer #1: NA

Reviewer #2: 1. Checking the association of sociodemographic characteristics for each individual NCD and communicable disorders is not required as per the objectives of the study and there is no need for authors to present these results as the numbers for each individual disorders are less.

2. Authors are requested to address the comments given during previous revision

**Do you want your identity to be public for this peer review?** For information about this choice, including consent withdrawal, please see our Privacy Policy..

Reviewer #1: No

Reviewer #2: No
